# Effects of Florfenicol on *nirS*-Type Denitrification Community Structure of Sediments in an Aquatic Microcosm Model

**DOI:** 10.3390/antibiotics12081254

**Published:** 2023-07-30

**Authors:** Tengyue Zhang, Junying Sun, Jinju Peng, Yuexia Ding, Yang Li, Haotian Ma, Mengbo Yu, Yi Ma

**Affiliations:** 1Department of Veterinary Medicine, College of Coastal Agricultural Sciences, Guangdong Ocean University, Zhanjiang 524088, China; zhangtengyue@stu.gdou.edu.cn (T.Z.);; 2College of Veterinary Medicine, Yangzhou University, Yangzhou 225000, China; 3Institute of Animal Health, Guangdong Academy of Agricultural Sciences, Guangzhou 510000, China

**Keywords:** florfenicol, *nirS* gene, denitrification community structure, sediment, aquatic microcosm

## Abstract

Florfenicol is one of the most widely used antibiotics in aquaculture and veterinary clinics because of its low side effects and strong bactericidal effect. A total of 45~60% of florfenicol is not absorbed by the animal body and accumulates in the aquatic environment through a variety of pathways, which affects denitrification. Indoor aquatic microcosm models were constructed and sediment samples were collected at different florfenicol concentrations (0.1, 1, 10, and 100 mg/L) on days 0, 7, 30, and 60 to extract the microbial genome DNA and determine the water properties. qPCR and amplicon sequencing were used to study the dynamic changes in the *nirS* gene and *nirS*-type denitrification community structure, diversity, and abundance, respectively. The results showed that high florfenicol stress influenced the sediment’s physicochemical properties, reducing conductivity, alkaline dissolved nitrogen, and organic matter content. In addition, the abundance of *nirS*, a functional denitrification gene, increased obviously with increased florfenicol concentrations but decreased the diversity of *nirS*-type denitrification microorganisms. Proteobacteria was the dominant denitrifying phylum in the sediment. Our study provides a scientific basis for the rational use of florfenicol in aquaculture to maintain a healthy and stable microecological environment and also provides a preliminary understanding of the response characteristics of water denitrifying microorganisms to florfenicol exposure.

## 1. Introduction

The use of antibiotics in livestock farming will continue to grow as the scale of intensive farming develops and expands, and the dependence on antibiotics in the farming process is becoming more severe. Global antibiotic consumption is expected to increase by 67% from 2010 to 2030 [1]. The annual usage of antibiotics in livestock farming globally is estimated to be 99,502 tons in 2020 and is expected to increase by 8% to 107,472 tons by 2030 [2]. However, only limited amounts of antibiotics are absorbed and utilized by human and animal organisms; most of the antibiotics are excreted in urine or feces [3,4]. The excreted antibiotics are released into the environment through direct or indirect ways and eventually pool in the aquatic environment. At present, antibiotic residues have been detected in various environments, including lakes, oceans, soils, sediments, farming wastewater, groundwater, surface water, and even drinking water. The detected concentrations of antibiotics were mostly in the ng or μg range in the environment, but the direct discharge of untreated livestock manure and farm wastewater can even lead to detected concentrations of antibiotics at the mg level in the surrounding environment, which seriously destabilizes the ecological environment [5,6,7]. In Korea, florfenicol was detected in the mainstream of the Han River at a concentration of 340 ng/L [8]. In Brazil and France, the highest concentrations of florfenicol have been reported in surface waters in the amount of 430 and 930 ng/L, respectively [9,10]. The residual concentration of florfenicol was as high as 11 mg/L in the waters around the Dalian Bay aquacultural farms in China [11]. In addition, Wang et al. examined the antibiotic content of children’s urine in Shanghai and found that florfenicol showed a frequency of detection of 32.3%, with a concentration of 67.71 μg/L [12].

Denitrification is an enzymatic reduction reaction mediated by microorganisms that use NO_3_^−^ or NO_2_^−^ as substrates to eventually produce N_2_O or N_2_, resulting in denitrification, which can occur in anaerobic, anoxic, and aerobic conditions [13,14,15]. Denitrification is the most important denitrification mechanism in freshwater lake systems, and its denitrification efficiency is much higher than anaerobic ammonia oxidation [16]. Nitrite reductases, coded by *nirK* and *nirS* genes together, are rate-limiting enzymes of the denitrification process that have the same function, but their subunit structures and numbers, catalytic sites, cofactors, and phylogenetic relationships are different [17]. Related studies have shown that *nirS*-type denitrification communities and *nirK*-type denitrification communities have different ecological niche preferences and response mechanisms to environmental factors [18,19]. Wang et al. [20] found that *nirK*-type denitrification communities have higher richness and diversity in soils [21]. Shi et al. [22] found that the *nirS*-type denitrification communities exhibited markedly more richness and diversity than *nirK*-type denitrification communities in the sediments of the southern bay of China. Some other studies have shown a correlation between antibiotic resistance genes and nitrogen cycling under antibiotic stress [23]. Antibiotics can inhibit nitrate and nitrite reduction by affecting key genes for denitrification [24], reducing functional enzyme activity [25], impeding the growth of denitrification bacteria [26,27], and altering the structure and diversity of the microbial community [28].

Florfenicol is one of the widely used antibiotics in the aquaculture industry, with annual usage once exceeding 10,000 tons [29,30], but there are only limited reports about the effects of florfenicol on the denitrification microbial community structure in aquatic ecosystems. In this study, the indoor construction of aquatic microcosm models was used to simulate aquatic ecosystems. The effects of florfenicol stress on the *nirS*-type denitrification microbial community structure in sediments were investigated by qPCR and high-throughput sequencing techniques to provide a preliminary understanding of the response characteristics of *nirS*-type denitrification microbes in sediment to florfenicol stress; to provide an important basis for an in-depth understanding of the effects of florfenicol on the environmental microbial community; and to provide the basis for better assessment and management regarding the environmental risk of florfenicol in the environment.

## 2. Results

### 2.1. Effects of Florfenicol Stress on Physicochemical Properties in Sediments

The physicochemical properties of the sediments are shown in Table 1. Sediment samples were all weakly alkaline, and florfenicol had no obvious effect on their pH. The conductivity of the sediment gradually decreased with increasing florfenicol concentration in the experimental groups. In control and low-concentration experimental groups, conductivity increased at 7 and 30 days followed by a decrease at 60 days in sediments. The conductivity under high-drug-concentration stress decreased at both 7 and 30 days, and the lowest value was reached at 30 days in the S4 group, with a marked rebound at 60 days. This indicates that high florfenicol concentration decreases the conductivity of sediments, and the conductivity gradually recovers with time. The alkaline nitrogen content of sediments gradually increased from 0 to 30 days; at 60 days, it increased in low-concentration experimental groups and decreased in control and high-concentration experimental groups. The organic matter content in the control groups showed an increasing trend; that in the experimental groups showed a decreasing trend, and at 60 days, the organic matter content was lower than at day 0 in all experimental groups.

### 2.2. Effects of Florfenicol on nirS Gene Abundance

The dynamic changes in *nirS*, the coding gene for nitrite reductase, in sediments were detected by qPCR, and results are shown in Figure 1. The absolute abundance of the *nirS* gene ranged from 3.59 × 10^4^ to 6.29 × 10^5^ copies/μL in sediments. In control groups (0 d and 0 mg/L), *nirS* gene abundance did not change markedly over time. In experimental groups, *nirS* gene abundance increased with increased drug concentration and time and reached the maximum *nirS* gene abundance at 60 days.

### 2.3. Effect of Florfenicol on nirS-Type Denitrification Community Structure

After the quality optimization of the original data, the number of valid sequences obtained in sediments was 91,816~189,063. After OTU cluster analysis, the number of OTUs in each sample ranged from 2420 to 5389, with an average of 3902 OTUs per sample. The 20 samples were divided into different groups according to the sampling time and florfenicol addition concentration, and then OTU clustering analysis was performed. According to the OTU cluster analysis results, a Venn diagram was drawn. The sampling time obviously decreases the number of co-OTUs in sediments; as shown in Figure 2, the number of co-OTUs in samples at 0 days was 1868; the number of co-OTUs in samples at 7, 30, and 60 days decreased sequentially to 1464, 686, and 580, respectively.

The microbial community diversity indices mainly include the Shannon and Simpson indices, and the richness indices mainly include the Chao1 and ACE indices. Larger indices represent a greater total species number and higher community diversity. Goods coverage is the sequencing depth index, which indicates the coverage of each sample library, whose higher values indicate the lower probability that sequences are not detected in the samples. As shown in Table 2, each sample had a Goods coverage index greater than 0.94, indicating that the *nirS* gene sequence detection coverage was high in samples, which could reflect the *nirS*-type denitrification microorganisms in sediments more realistically and comprehensively. At 7 days, florfenicol stress did not affect the Shannon index for the denitrification microbial community; at 30 days, the Shannon index decreased in all experimental groups, and it decreased with increased florfenicol concentration at 60 days. The richness index of the denitrification microbial community showed an increasing trend at 7 days and decreased obviously at 30 days; at 60 days, the richness index decreased continuously in high-concentration experimental groups. OTUs numbers showed a similar trend to the richness index, and the OTUs number decreased by 30.18% and 42.43% in high-concentration experimental groups at 60 days, respectively.

The *nirS*-type denitrification microbial community composition of each sample was statistical under different species’ classification levels. The identified species were classified into 7 phyla 12 classes, 18 orders, 21 families, 41 genera, and 81 species. The relative abundance of each species in the denitrifying communities was calculated, with the relative abundance of unclassified species ranging from 74.35% to 87.07%. At the phylum level, the relative abundance of *Proteobacteria* ranged from 5.10% to 24.49% and mainly included *Betaproteobacteria*, *Alphaproteobacteria*, and *Gammaproteobacteria*, which were the absolute dominant bacteria. The relative abundance of *Betaproteobacteria* was the highest and the relative abundance of *Gammaproteobacteria* was the lowest. The top 30 species at the genus level were selected to draw a clustering heat map, and the similarities and differences in species distributions were visualized by their color differences, as shown in Figure 3. *Thauera*, *Azoarcus*, *Dechloromonas*, *Acidovorax*, *Thiobacillus*, *Comamonas*, *Pseudomonas*, and *Rhodanobacter* were the dominant bacteria at the genus level, with relative abundance more than 0.5%. Some bacteria showed obvious changes in relative abundance, among which the relative abundance of *Thauera* in samples S3D60 and S4D60 was only 0.94% and 1.2%, with an obvious decrease compared to the other groups; in sample S4D60, the relative abundance of *Azoarcus*, *Dechloromonas,* and *Alicycliphilus* was 8.93%, 6.15%, and 3.89%, respectively, which clearly increased compared with the other groups. The relative abundance of *Marinobacter* in S3D60 was 5.62%, but the relative abundance in the other groups was less than 0.3%.

### 2.4. Beta Diversity

The PCoA (Principal coordinates analysis) and UPGMA (unweighted pair group method with arithmetic mean) clustering methods were used to express the similarity and difference degrees between different samples. As shown in Figure 4, the contributions of PC1 and PC2 principal coordinates to sample differences were 20.45% and 16.96%, respectively, in PCoA. Samples with different sampling times were completely separated, indicating that sampling time was the dominant factor in the degree of variation between samples. The treatment groups with higher florfenicol addition concentrations at 30 and 60 days were the furthest away from the control group, indicating that the *nirS*-type denitrification microbial community was more different and more influenced by the drug. Similarly, treatment groups with 10 and 100 mg/L florfenicol added concentrations at 30 and 60 days clustered together and were the furthest away from the control group in the cluster analysis. Samples with the same sampling time were clustered together in the cluster analysis. The cluster analysis results were consistent with the dispersion between samples in the PCoA analysis.

### 2.5. Linkage among Sediment Properties, nirS Gene Abundance, and nirS-Type Denitrification Communities

The top 10 *nirS*-type denitrification bacteria with higher relative abundance in each sample were selected, and redundancy analysis (RAD) was used to investigate their correlations with sediment properties and *nirS* gene abundance. The results are shown in Figure 5. *Dechloromonas*, *Marinobacter,* and *Azoarcus* showed clearly positive correlations with *nirS* genes; *Thiobacillus*, *Thauera,* and *Rhodanobacter* showed clearly negative correlations with *nirS* genes. *Thiobacillus* showed the strongest correlation with pH, with a clearly positive correlation. *Pseudomonas*, *Alicycliphilus*, *Acidovorax,* and *Comamonas* showed clearly positive correlations with organic matter and negative correlations with alkaliolytic nitrogen. *Thauera* and *Rhodanobacter* showed positive correlations with electrical conductivity and alkaliolytic nitrogen.

## 3. Discussion

Antibiotic residues in the environment cause widespread dissemination and prevalence of antibiotic resistance genes and also affect the nitrogen cycle, especially the denitrification process in aquatic systems, which is important for mitigating nitrate pollution. Some studies have reported that antibiotics affect or inhibit the denitrification process to varying degrees [31,32,33], inhibiting the activity of key functional enzymes of denitrification microorganisms [32]. However, it has also been reported that the antibiotic negative effects decrease to a degree over time and a gradual adaptation to antibiotic stress can be observed overall [33,34]. Wu et al. [35] found that denitrification functional gene levels were reduced under sulfadimethoxine stress and that *Pseudomonas* was sufficiently enriched as a major antibiotic resistance genes carrier. Sun et al. [36] found that denitrification significantly reduced antibiotic levels and antibiotic resistance gene abundance, and there was a significant negative correlation between denitrification functional genes and antibiotic resistance genes. However, some studies also reported that trace and low concentrations of antibiotics (spiramycin, streptomycin, hygromycin, and ciprofloxacin) did not significantly inhibit denitrification efficiency but resulted in reduced abundance of related bacterial genera and enrichment of antibiotic resistance genes [37,38]. Some antibiotics also promoted the proliferation of denitrifying bacteria and partial nitrification and denitrification to improve nitrogen pollution [39,40]. Gonzalez et al. [41] found that norfloxacin and sulfamethoxazole significantly increased the expression of denitrification genes and antibiotic resistance genes. Li et al. [42] and Huang et al. [43] found that tetracycline promoted the growth of denitrifying bacteria. Roose-Amsaleg et al. [44] found that tetracycline at therapeutic concentrations (10 mg/L) altered the community structure of denitrifying bacteria but did not affect the denitrification process, while Zhang et al. [45] observed that the accumulation of nitrate and nitrite was 4.21 and 10 times higher, respectively, than the control after exposure to 2 mg/kg tetracycline in the same sediment. An et al. [46] found that an increased concentration of sulfamethoxazole from 0.1 to 10 mg/L significantly inhibited nitrate removal in the wastewater. However, Fan et al. [47] found that low concentrations (<0.2 mg/L) of sulfamethoxazole led to a decrease in nitrate removal in earlier stages, but higher concentrations (0.4 to 20 mg/L) of sulfamethoxazole had no further inhibitory effect. Obviously, there are different conclusions on the effects of antibiotics on denitrification and even some contradictory conclusions. This may be related to the composition of denitrification microorganisms and the sensitivity of different denitrification bacteria to antibiotics.

In our study, high florfenicol stress influenced sediment physicochemical properties, reducing conductivity, alkaline-dissolved nitrogen, and organic matter content. In addition, the abundance of *nirS*, a functional denitrification gene, increased obviously with increased florfenicol concentrations but decreased the diversity of *nirS*-type denitrification microorganisms. Several studies have shown that florfenicol slightly promotes nitrite reductase encoding gene abundance for a short period of time [32,48]. The same phenomenon was found in our previous study: the florfenicol addition to aquatic microcosms suppressed the aquatic denitrifying microbial community diversity but increased the *nirS* gene abundance, and in addition, the treatment group with lower florfenicol addition induced a marked increase in florfenicol-resistant genes in aquatic environments [31,49]. This may be due to the combined effect of drug and time that some strains carrying *nirS* genes acquired florfenicol resistance genes and became dominant bacteria under florfenicol stress. However, functional gene abundance only indicates the potential for functional activity, and the actual role may be limited by mRNA expression and post-transcriptional modifications of functional genes in the environment [50]. Certain bacteria can adapt to poorer survival environments by increasing their transcriptional activity [51]. It was found that functional genes for denitrification processes in marine and estuarine environments have low, and in some cases undetectable, transcriptional abundance [52,53]. Exposure to high antibiotic concentrations may stimulate the transfer and replication of plasmids carrying resistance genes in denitrifying microbial communities, causing resistance gene abundance to increase; in addition, denitrifying bacteria may develop resistance to antibiotics to adapt to antibiotic-induced stress.

## 4. Materials and Methods

### 4.1. Reagents

The lake surface water (1–10 cm) and lake surface sediments (1–10 cm) in Guangdong Ocean University (Zhanjiang, China) were collected, and the debris was removed to construct an aquatic microcosm. Florfenicol (13021322) was purchased from North China Pharmaceutical Co., Ltd. (Shijiazhuang, China). The Soil DNA Kit (D5625) and MicroElute Gel Extraction Kit (D6294) were purchased from Omega Bio-Tek Company (Guangzhou, China). The TIANprep Midi Plasmid Kit (DP106-02) was purchased from Tiangen Biochemical Technology Co., Ltd. (Beijing, China). The pMD^TM^19-T Vector Cloning Kit (6013) was purchased from Bao Bioengineering Co., Ltd. (Dalian, China). The ChamQ Universal SYBR qPCR Master Mix (Q711- 02) was purchased from Nanjing Novizan Biotechnology Co., Ltd. (Nanjing, China). The DH5α competent cells (B528413), Real-time PCR Plates, 96-Well Transparent, Non Skirted (F603101), and primers were purchased from Sangon Bioengineering Co., Ltd. (Shanghai, China).

### 4.2. Aquatic Mesocosm Experiment Design and Sampling

The mesocosm experiment was constructed to simulate the aquatic ecosystem. Surface water (1–10 cm) and surface sediments (1–10 cm) were collected from lakes at the Guangdong Ocean University (Zhanjiang, China), larger stones and other debris were removed, and the samples were divided into transparent plastic boxes (50 × 40 × 30 cm). Each plastic box contained 30 L of lake water and 10 cm of sediment, and the depth of water was 15 cm. The constructed aquatic mesosystems were stabilized at room temperature (25 ± 3 °C) for 3 days. Florfenicol solution was added so that the concentration of florfenicol in the water column was 0 (control group), 0.1, 1, 10, and 100 mg/L. Among them, 0 mg/L was the control group, 0.1 and 1 mg/L were the low-concentration experimental groups, and 10 and 100 mg/L were the high-concentration experimental groups. Sediment samples (numbered S0 to S4) were collected on days 0, 7, 30, and 60 (D0, D7, D30, and D60; among them, D0 was the control group). The in vitro bacteriostatic effect of the configured florfenicol solution was tested using the microbroth twofold dilution method, and the MIC of florfenicol against Escherichia coli was found to be 8 μg/mL, which ensured that the solution had a significant bacteriostatic effect. The experiment had 5 groups with 3 replicates each. Each group of three samples was divided into two parts, one of which was used for the determination of physical and chemical properties; the other part was used for DNA extraction, and the DNA was mixed before being sent for sequencing.

### 4.3. Analysis of Sample Physical and Chemical Properties

Several experiments were performed to assess the physiochemical characteristics of the samples. For example, sediment pH was measured using a pH meter (PHS-3B, Leici Shanghai, Shanghai, China ) with a water-to-soil ratio of 2.5:1; sediment conductivity was measured using a conductivity meter (DDS-307A, Leici Shanghai) with a water-to-soil ratio of 5:1; the volumetric method (using potassium dichromate) was used to measure the amount of organic matter; and alkaline hydrolysis was used to determine the amount of alkali-hydrolyzable nitrogen. Each sample was analyzed in triplicate.

### 4.4. DNA Extraction and qPCR

According to the manufacturer’s instructions, genomic DNA was extracted from sediment samples using the soil DNA kit (Omega Bio-Tek, Norcross, GA, USA). The concentration and purity of the extracted DNA were assessed using a nanodrop UV–vis spectrophotometer. Three DNA samples were extracted from each group for analysis. Based on previous work, the SYBR-primer method was employed with the CFX Connect Real-Time System instrument (BIO-RAD, Hercules, CA, USA) to determine the abundance of the *nirS* gene. The denitrifying bacterial *nirS* gene was amplified using the primer pair CD3AF/R3CDR. The qPCR reaction system was as follows: 10 µL SYBR, 1 µL DNA template, 1 µL upstream and downstream primers (10 mol·L^−1^) each, and 8 µL ddH2O. The qPCR reaction program was as follows: 94 °C, 5 min; 94 °C for 30 s, 58 °C for 30 s, 72 °C for 30 s, 39 cycles. The absolute abundance of *nirS* genes was calculated according to the method of Zhang et al. [31].

### 4.5. Amplicon Sequencing of nirS Gene and Bioinformatics Processing

The DNA samples from each of the three replicates were combined into the mixed sample and sent to Jinweizhi (Suzhou, China) for Illumina MiSeq paired-end sequencing of the *nirS* gene amplicon regions. The *nirS* gene was amplified for sequencing analysis using the same primer pairs as qPCR. The high-throughput sequencing raw data were optimized using QIIME 1.9.1 by splicing overlapping regions at the ends of the sequences, removing sequences that were shorter than 200 bp, and removing chimeric sequences to produce valid data. An operational taxonomic unit (OTU) cluster analysis was carried out on the basis of a 97% similarity. The species taxonomic annotation was completed using the NCBI database. Based on OTU analysis, results were obtained using the method of random sampling; sample sequences were flat, and we calculated the Shannon and Chao1 alpha diversity indices, community species abundance, and diversity. The diversity index was examined using QIIME 1.9.1 software and R language.

### 4.6. Statistical Analysis

Excel 2016 was used to analyze the *nirS* gene abundance, QIIME 1.9.1 software was used to perform the α diversity index analysis, R language was used for β diversity analysis, and the correlation between environmental factors and community structure was examined using CANOCO 5.0 software.

## 5. Conclusions

In this study, *Proteobacteria* was the dominant denitrifying phylum under florfenicol stress in sediment. High florfenicol stress influenced sediment physicochemical properties, reducing conductivity, alkaline-dissolved nitrogen, and organic matter content. In addition, the abundance of *nirS*, a functional denitrification gene, increased obviously with increased florfenicol concentrations but decreased the diversity of *nirS*-type denitrification microorganisms. Proteobacteria was the dominant denitrifying phylum in sediment.

## Figures and Tables

**Figure 1 antibiotics-12-01254-f001:**
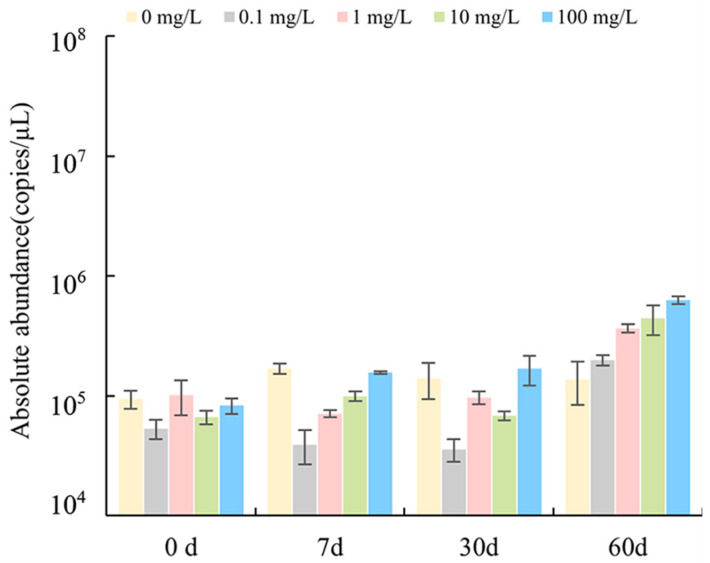
Effects of different concentrations of florfenicol on the abundance of *nirS* genes in sediments.

**Figure 2 antibiotics-12-01254-f002:**
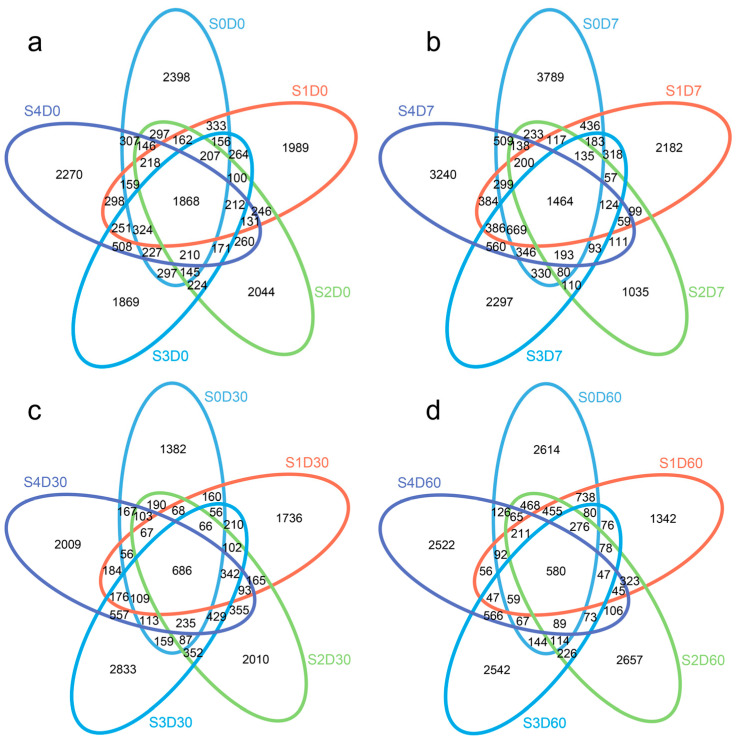
Effects of different concentrations of florfenicol on operational taxonomic units (OTUs) of bacteria (Venn diagram) in sediments. Effects of different concentrations of florfenicol on OTUs at 0 days (**a**), 7 days (**b**), 30 days (**c**), and 60 days (**d**). Different colors of circles represent different samples, and the numbers represent the number of unique OTUs in each sample or the number of common OTUs in all samples.

**Figure 3 antibiotics-12-01254-f003:**
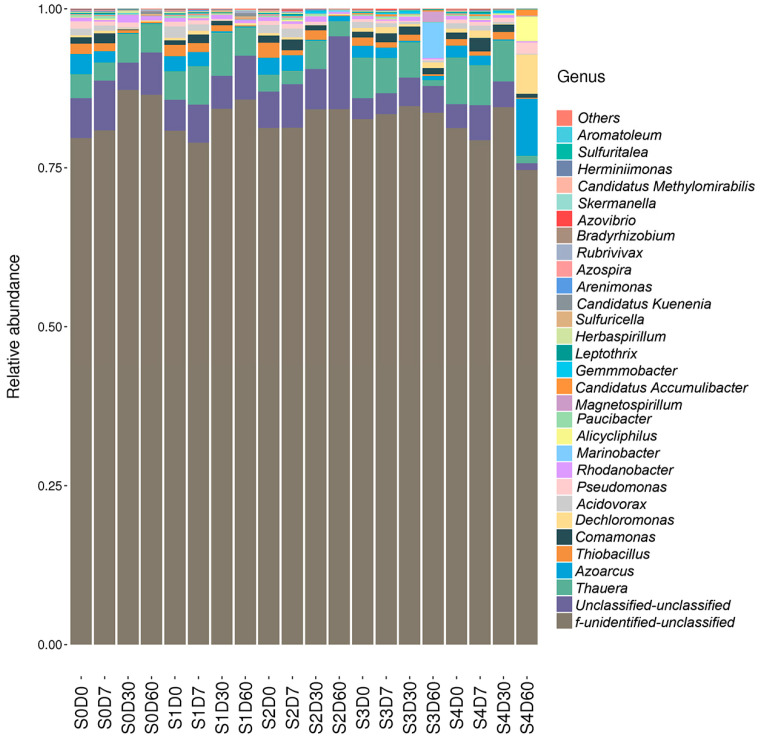
Effects of different concentrations of florfenicol on genus-level species abundance on *nirS*-type denitrification microbes in sediments.

**Figure 4 antibiotics-12-01254-f004:**
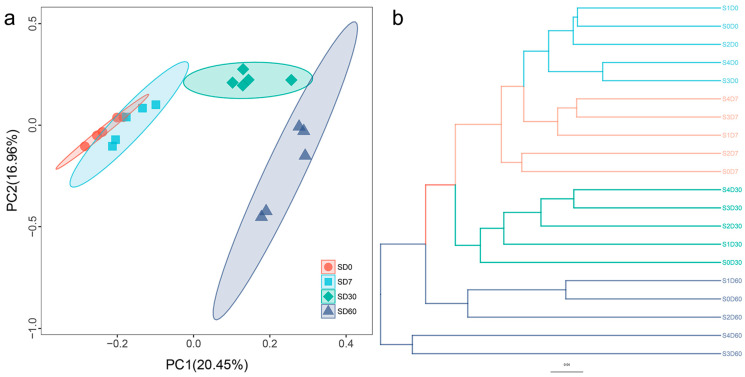
Principal coordinates analysis (PCoA) and unweighted pair group method with arithmetic mean (UPGMA) reveal the degrees of similarity and difference of the *nirS*-type denitrification microbial community between different sediment samples. (**a**) The results of the PCoA analysis. (**b**) The results of the UPGMA analysis.

**Figure 5 antibiotics-12-01254-f005:**
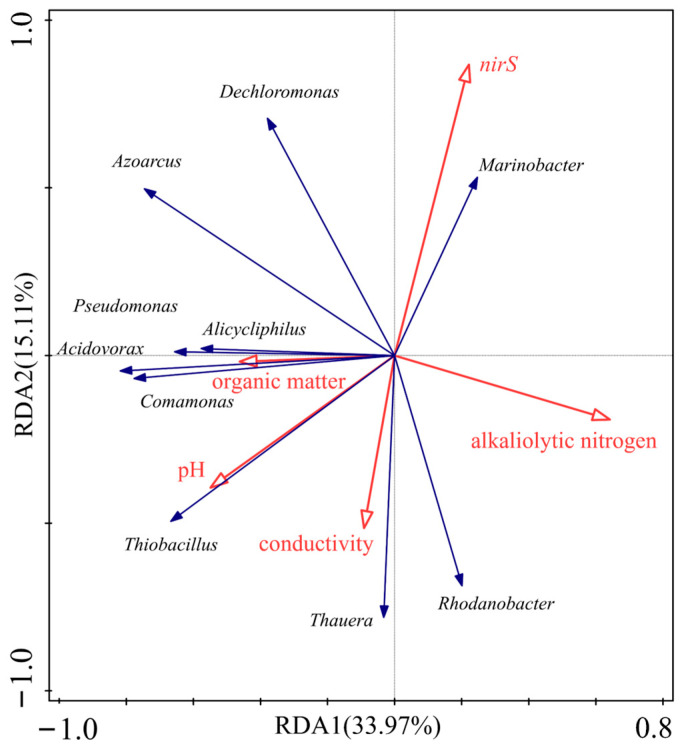
Correlation analysis between *nirS*-type denitrification microbial and sediment properties and *nirS* gene abundance in aquatic microcosm models.

**Table 1 antibiotics-12-01254-t001:** Physicochemical properties in sediments.

Sample	pH	Conductivity (μS/cm)	Alkaline Nitrogen	Organic Matter
			(mg/kg)	(g/kg)
S0D0	7.88 ± 0.03	208.00 ± 5.29	173.42 ± 1.03	21.76 ± 0.51
S1D0	7.99 ± 0.12	221.00 ± 2.65	187.03 ± 2.20	26.46 ± 1.53
S2D0	7.88 ± 0.03	213.00 ± 5.29	191.67 ± 12.21	30.09 ± 0.57
S3D0	8.04 ± 0.12	204.00 ± 4.00	179.68 ± 5.06	26.55 ± 0.69
S4D0	7.64 ± 0.02	219.00 ± 2.65	196.89 ± 1.57	33.41 ± 0.62
S0D7	7.99 ± 0.04	249.00 ± 6.08	202.72 ± 1.66	24.27 ± 2.92
S1D7	8.01 ± 0.03	256.00 ± 3.61	199.67 ± 4.02	25.33 ± 2.20
S2D7	7.81 ± 0.08	251.00 ± 5.20	223.41 ± 4.72	30.43 ± 1.10
S3D7	7.96 ± 0.05	233.00 ± 5.57	209.62 ± 4.39	27.10 ± 0.56
S4D7	7.63 ± 0.04	202.00 ± 7.94	216.16 ± 1.99	28.75 ± 1.32
S0D30	8.03 ± 0.04	261.00 ± 9.54	313.54 ± 9.12	20.76 ± 0.51
S1D30	8.00 ± 0.03	275.00 ± 9.17	273.53 ± 6.99	24.20 ± 1.48
S2D30	7.78 ± 0.06	274.00 ± 4.36	261.63 ± 4.63	28.38 ± 0.90
S3D30	7.97 ± 0.03	228.00 ± 3.61	267.23 ± 3.66	25.49 ± 0.57
S4D30	7.80 ± 0.04	119.00 ± 4.58	339.91 ± 4.98	30.68 ± 2.18
S0D60	7.50 ± 0.03	215.00 ± 1.73	248.44 ± 4.99	23.03 ± 1.26
S1D60	7.64 ± 0.02	205.00 ± 9.54	300.01 ± 6.86	24.13 ± 0.66
S2D60	7.56 ± 0.05	195.10 ± 2.01	275.04 ± 2.65	29.45 ± 1.16
S3D60	7.77 ± 0.08	170.80 ± 6.58	225.58 ± 4.84	22.13 ± 0.76
S4D60	7.68 ± 0.06	180.80 ± 4.53	268.63 ± 6.06	29.66 ± 1.30

**Table 2 antibiotics-12-01254-t002:** Alpha diversity index of *nirS*-type denitrification microbial in sediments.

Sample	OTUs	Chao1	Ace	Shannon	Simpson	Goods Coverage
S0D0	4592	10,247	10,985	10.69	1.00	0.95
S1D0	4312	9601	10,055	10.60	1.00	0.96
S2D0	4020	9680	10,065	10.41	1.00	0.96
S3D0	4626	8946	9474	10.38	1.00	0.96
S4D0	4879	9715	10,414	10.52	1.00	0.96
S0D7	5192	13,508	14,433	11.13	1.00	0.94
S1D7	4389	10,180	10,461	10.72	1.00	0.96
S2D7	3161	5200	5288	10.32	1.00	0.98
S3D7	4678	9621	10,170	10.34	1.00	0.96
S4D7	5389	11,611	12,638	10.78	1.00	0.95
S0D30	2420	4662	5190	9.31	1.00	0.98
S1D30	2636	5766	6551	9.64	1.00	0.97
S2D30	3374	6993	7820	9.85	1.00	0.97
S3D30	4188	8338	9154	10.20	1.00	0.96
S4D30	3921	6773	7443	9.91	1.00	0.97
S0D60	3796	8139	9047	9.67	0.99	0.96
S1D60	2982	5952	6233	9.71	1.00	0.98
S2D60	3455	7962	8615	8.73	0.98	0.96
S3D60	3230	6339	6962	8.29	0.98	0.97
S4D60	2809	6447	7130	7.12	0.94	0.97

## Data Availability

The 16S rDNA sequencing data have been uploaded to the repository (National Center for Biotechnology Information, NCBI), which can be found at the link http://www.ncbi.nlm.nih.gov/bioproject/986876, and the Accession no. is PRJNA986876, accessed on 23 July 2023.

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
