# Peer review of "Effects of Florfenicol on nirS-Type Denitrification Community Structure of Sediments in an Aquatic Microcosm Model"

_antibiotics, 2023, doi:10.3390/antibiotics12081254_

Round 1
Reviewer 1 Report
Antibiotics 2494966.v1
The manuscript is very interesting, having considerable data for potential publication. Still for my comprehension and manuscript’s improvement, I recommend the following for authors’ consideration during next phase of the publication:
The abundance of nirS, a functional denitrification gene, increased with increased florfenicol concentrations but decreased the diversity of nirS-type denitrification microorganisms; so how the observed nirS increase is justified? The above scenario may indicate the selection of some microbes having nirS gene; in that case what could be the factors/strategies for such potential selection?
An increase in organic matter, during experimental incubation, was observed with no change in pH; while the OM should produce carboxylic acids for normal ecosystems, to ultimately decrease pH though slight change. Please describe the phenomenon. Moreover, increased OM should also facilitate microbial community diversity.
As per section 4.2, microcosms were stabilized for 3-days; what does it mean? And which factors the authors considered for this stabilization?
As the denitrification process is mainly anaerobic, so the depth of water in microcosms is very important; the authors have mentioned the dimensions of the boxes (30.40.50 cm), bu the depth of water would be better to mention for clear understanding.
Please correct et al throughout the manuscript as per guidelines of the journal.
At several places there is no space between brackets and the words, see e.g., L58; please correct throughout the manuscript.
The authors have quite a data while their discussion needs further flesh accordingly. For instance the antibiotics degradation and the impacts of potential metabolites on nitrogen cycle; and, enriched discussion of ‘physicochemical parameter studied here and their inter-relationships with nirS abundance and harboring community.
L49: correct abbreviation for nitrates and nitrites.
L70: change the word ‘experiment’ with study.
L245-247: conclusion statement like this should be avoided; authors should conclude something on the basis of their results and not be part of the scientific ambiguity.
Please improve the figure quality / readability for Fig 4b.
The language looks fine.
Author Response
The manuscript is very interesting, having considerable data for potential publication. Still for my comprehension and manuscript’s improvement, I recommend the following for authors’ consideration during next phase of the publication:
-The abundance of nirS, a functional denitrification gene, increased with increased florfenicol concentrations but decreased the diversity of nirS-type denitrification microorganisms; so how the observed nirS increase is justified? The above scenario may indicate the selection of some microbes having nirS gene; in that case what could be the factors/strategies for such potential selection?
>>>> Thanks very much for the reviewer’s valuable comments and suggestions. We explored the diversity and richness of nirS-type denitrifying microorganisms in sediments by amplicon sequencing, and quantified the absolute abundance of nirS genes by qPCR, and the result showed that the nirS gene abundance increased with florfenicol concentration, but the diversity of nirS-type denitrifying microorganisms decreased. This is perhaps due to that certain nirS-type denitrifying microorganisms are resistant to florfenicol and are able to grow normally under florfenicol stress, thus being selected and enriched, thanks again.
-An increase in organic matter, during experimental incubation, was observed with no change in pH; while the OM should produce carboxylic acids for normal ecosystems, to ultimately decrease pH though slight change. Please describe the phenomenon. Moreover, increased OM should also facilitate microbial community diversity.
>>>> Thanks very much for the reviewer’s valuable comments and suggestions. As organic matter increases, decomposing microorganisms metabolize the organic matter and produce some acidic metabolites, such as organic acids. These organic acids can decrease aquatic pH. However, it should be noted that effects of increased organic matter on pH are also regulated by other factors, such as the alkaline content and the physicochemical conditions of water and sediment, etc. Our previous study showed that florfenicol stress caused an increase in pH of water bodies from 7.83 to 9.05, which also affects pH in the sediment. In summary, although increased organic matter can have some effects on the pH of water bodies, the exact extent of the effect will be moderated by other factors, thanks again.
-As per section 4.2, microcosms were stabilized for 3-days; what does it mean? And which factors the authors considered for this stabilization?
>>>> Thanks very much for the reviewer’s valuable comments and suggestions. We collected lake water bodies and sediments to remove larger impurities and mix them well, and then dispensed them in 15 aquatic microcosm models. Some of the sediments were suspended in the water column due to violent agitation during the partitioning process of the water and sediments, causing turbidity in water bodies. The aquatic microcosm model was placed at room temperature for 3 days, and it was observed that the water body became clear and transparent again, at this time, we thought that the heavier bottom sediment particles had been deposited to the bottom, and that the micro-ecosystem was basically stabilized, thanks again.
-As the denitrification process is mainly anaerobic, so the depth of water in microcosms is very important; the authors have mentioned the dimensions of the boxes (30.40.50 cm), bu the depth of water would be better to mention for clear understanding.
>>>> Thanks very much for the reviewer’s valuable comments and suggestions. The plastic boxes used to construct the aquatic microcosm model had the size of 50 × 40 × 30 cm (length × width × height), with 30 L water in each box and the depth of water was 15 cm. We have supplemented the information on line 52 in manuscript, thanks again.
-Please correct et al throughout the manuscript as per guidelines of the journal.
>>>> Thanks very much for the reviewer’s valuable comments and suggestions. We have revised the manuscript in accordance with journal requirements and reviewer comments, thanks again.
-At several places there is no space between brackets and the words, see e.g., L58; please correct throughout the manuscript.
>>>> Thanks very much for the reviewer’s valuable comments and suggestions. We have revised the manuscript in accordance with journal requirements and reviewer comments, thanks again.
-The authors have quite a data while their discussion needs further flesh accordingly. For instance the antibiotics degradation and the impacts of potential metabolites on nitrogen cycle; and, enriched discussion of ‘physicochemical parameter studied here and their inter-relationships with nirS abundance and harboring community.
>>>> Thanks very much for the reviewer’s valuable comments and suggestions. We have revised the discussion and added sections to the manuscript, thanks again.
-L49: correct abbreviation for nitrates and nitrites.
>>>> Thanks very much for the reviewer’s valuable comments and suggestions. We have revised the manuscript in accordance with reviewer comments, thanks again.
-L70: change the word ‘experiment’ with study.
>>>> Thanks very much for the reviewer’s valuable comments and suggestions. We have revised the manuscript in accordance with reviewer comments, thanks again.
-L245-247: conclusion statement like this should be avoided; authors should conclude something on the basis of their results and not be part of the scientific ambiguity.
>>>> Thanks very much for the reviewer’s valuable comments and suggestions. We fully agree with you that similar conclusion statements should be avoided. We have amended and improved this section as a part of the Discussion section, in addition we have added the Conclusion section and ensured that the Conclusion section does not contain scientific ambiguities but is an accurate and reliable interpretation based on experimental results, thanks again.
-Please improve the figure quality / readability for Fig 4b.
>>>> Thanks very much for the reviewer’s valuable comments and suggestions. We modified Figure 4 to make the clustering of images clearer by distinguishing the differences between groups by color, thanks again.
>>>>In addition to these revisions suggested by reviewer, we also improved our manuscript by supporting some other changes which have marked in red color in the revised manuscript. We hope these changes and corrections could make this manuscript more precise.
>>>>We have tried our best to improve the manuscript and made some changes in the manuscript that were marked. These changes will not influence the framework of the manuscript. We appreciate for Reviewer’s warm work earnestly, and hope that the corrections will meet with approval.
Reviewer 2 Report
The results of this study showed that high florfenicol stress influenced sediment physicochemical properties, reducing conductivity, alkaline dissolved nitrogen, and organic matter content. In addition, the abundance of nirS, a functional denitrification gene, increased with increased florfenicol concentrations but decreased the diversity of 24 nirS-type denitrification microorganisms.
Line 131: florfenicol stress did not affect of Shannon index; change to: florfenicol stress did not affect Shannon index
Line 155: the relative abundance of… were; change to: the relative abundance of… was (singular)
Line 167+168: the contribution of… were; change to: the contribution od… was (singular)
Line 239: Zhang et al [45]; change to: Zhang et al [46]
Line 241: An and Qin [46]; change to: An and Qin [45]
Line 269: Keep at ambient temperature…….; rewrite!
Line 275-277: Three replicate samples from each group were mixed and immediately split into two parts for the determination of physicochemical parameters and DNA extraction. After this procedure are they real replicates??
Line 434: 45 An, Y.L.; Qin, X.M. not referenced in the text.
Some English grammar mistakes and some incomplete sentences
Author Response
The results of this study showed that high florfenicol stress influenced sediment physicochemical properties, reducing conductivity, alkaline dissolved nitrogen, and organic matter content. In addition, the abundance of nirS, a functional denitrification gene, increased with increased florfenicol concentrations but decreased the diversity of 24 nirS-type denitrification microorganisms.
-Line 131: florfenicol stress did not affect of Shannon index; change to: florfenicol stress did not affect Shannon index
>>>> Thanks very much for the reviewer’s valuable comments and suggestions. We have revised the manuscript in accordance with reviewer comments, thanks again.
-Line 155: the relative abundance of… were; change to: the relative abundance of… was (singular)
>>>> Thanks very much for the reviewer’s valuable comments and suggestions. We have revised the manuscript in accordance with reviewer comments, thanks again.
-Line 167+168: the contribution of… were; change to: the contribution od… was (singular)
>>>> Thanks very much for the reviewer’s valuable comments and suggestions. We have revised the manuscript in accordance with reviewer comments, thanks again.
-Line 239: Zhang et al [45]; change to: Zhang et al [46]
-Line 241: An and Qin [46]; change to: An and Qin [45]
-Line 434: 45 An, Y.L.; Qin, X.M. not referenced in the text.
>>>> Thanks very much for the reviewer’s valuable comments and suggestions. We apologized for getting the order of the two references wrong, we have reordered the two references in the reference list and changed the form of the reference in the draft to "An et al", thanks again.
-Line 269: Keep at ambient temperature…….; rewrite!
>>>> Thanks very much for the reviewer’s valuable comments and suggestions. We have modified this sentence to read “The constructed aquatic mesosystems were stabilized at room temperature (25±3°C) for 3 days”, thanks again.
-Line 275-277: Three replicate samples from each group were mixed and immediately split into two parts for the determination of physicochemical parameters and DNA extraction. After this procedure are they real replicates?
>>>> Thanks very much for the reviewer’s valuable comments and suggestions. The manuscript is not clear enough about how the samples are treated, so we have added the relevant content. "Each group of three samples is divided into two parts, one of which is used for the determination of physical and chemical properties, the other part is used for DNA extraction, and the DNA is mixed before being sent for sequencing." The determination of physical and chemical properties of the environment here are three real duplicate samples, while the amplicon sequencing uses mixed samples, thanks again.
>>>>In addition to these revisions suggested by reviewer, we also improved our manuscript by supporting some other changes which have marked in red color in the revised manuscript. We hope these changes and corrections could make this manuscript more precise.
>>>>We have tried our best to improve the manuscript and made some changes in the manuscript that were marked. These changes will not influence the framework of the manuscript. We appreciate for Reviewer’s warm work earnestly, and hope that the corrections will meet with approval.
Reviewer 3 Report
L. 33 – 40 ... there is no clarification as to whether the data is global or only concerns China. The use of ATB is strictly regulated in the EU, so it can be expected that the increase in residues in the environment will vary in the EU and rather it is expected to decrease.
L. 41-47 …. there is a lack of comparison of which types of environment are most affected. The text is too general and declarative.
L. 70... the reason why the model experiment was chosen, what advantages it has, is not clarified. What is the benefit of the model experiment; how much the results will correspond to the natural state and whether indeed florfenicol residues in nature can significantly affect microbial denitrification and in which environments this can be expected. Disadvantages should also be listed. There is a lack of information on the concentrations of florfenicol in different natural types of environments.
L.74... the effect of stress induced by florfenicol should have been supplemented with additional information regarding e.g. MIC MBC, which will have major impacts on the abundance and qualitative composition of the bacterial community.
L.75… The goal is unconvincingly worded. The expected main contribution of the so called "novelty" of the study is not specified.
L. 98…how is the control marked?
L.103… these are speculations, unsubstantiated results. Have resistance genes been studied?
L. 162... Fig.3 legend is not accurate. The colour range is bland.
L.172…. where specific results characterizing the claimed differences between the samples are given... "denitrification microbial community was more different between samples and more influenced by the drug"….
L.223-225... these are unconvincing conclusions. Have similar results been found with other ATBs? Citations need to be added. Model studies have also been carried out, this may play a significant role.
L. 245-248…. The final conclusion is contradictory. The specific contribution of the study is not clearly stated. The expected aim of the study, that the results provide a theoretical basis for the scientific use of florfenicol in aquaculture and the maintenance of a healthy aquaculture ecosystem....was not fulfilled, as it is not clear how florfenicol will realistically affect the natural processes of microbial denitrification in the natural environment.
L. 267-277… Design of the experiment is insufficiently clarified.
Why were concentrations of the florfenicol in the decimal series chosen? Is it not justified how the sampling intervals were chosen? Is florfenicol's natural half-life interval not listed? Are control samples anywhere detail characterized? Has a positive check been made with, for example, another ATB?
L- 279... devices are missing the manufacturer's indication.
Author Response
-L. 33 – 40 ... there is no clarification as to whether the data is global or only concerns China. The use of ATB is strictly regulated in the EU, so it can be expected that the increase in residues in the environment will vary in the EU and rather it is expected to decrease.
>>>> Thanks very much for the reviewer’s valuable comments and suggestions. The numbers in the manuscript are global antibiotic use, with "Global antibiotic consumption is expected to increase by 67% from 2010 to 2030" representing total global antibiotic use and "The annual usage of antibiotics in livestock farming globally is estimated to be 99,502 tons in 2020 and is expected to increase by 8% to 107, 472 tons up to 2030" representing global antibiotic use in animal agriculture, both of which are expected to increase, thanks again.
-L. 41-47 …. there is a lack of comparison of which types of environment are most affected. The text is too general and declarative.
>>>> Thanks very much for the reviewer’s valuable comments and suggestions. We added some concentrations of florfenicol residues in aquatic systems in manuscript and added relevant literature to make the presented text more specific. We added the following to the manuscript: “In Korea, florfenicol was detected in the main stream of the Han River at a concentration of 340 ng/L [8]. In Brazil and France, the highest concentrations of florfenicol have been reported in surface waters of 430 and 930 ng/L, respectively [9-10]. The residual concentration of florfenicol was as high as 11 mg/L in the waters around the Dalian Bay aquacultural farms in China [11]. In addition, Wang et al examined the antibiotic content of children's urine in Shanghai and found that FF showed a frequency of detection of 32.3%, with a concentration of 67.71 μg/L [12]”, thanks again.
-L. 70... the reason why the model experiment was chosen, what advantages it has, is not clarified. What is the benefit of the model experiment; how much the results will correspond to the natural state and whether indeed florfenicol residues in nature can significantly affect microbial denitrification and in which environments this can be expected. Disadvantages should also be listed. There is a lack of information on the concentrations of florfenicol in different natural types of environments.
>>>> Thanks very much for the reviewer’s valuable comments and suggestions. The aquatic microcosm model is an experimental tool for simulating ecosystems to study microbial interactions and ecological processes. The model is experimentally controllable, allows for repetitive experiments, provides reliable and stable experimental results, and facilitates observation and sampling, enabling effective access to information about the structure and function of microbial communities.
In our experiments, the water bodies and sediments were collected from natural lakes, which are relatively close to the actual situation in the natural environment. However, we strongly agree with the reviewers that the aquatic microcosm model cannot fully replicate the complexity and dynamics in the natural environment. Experimental results based on the aquatic microcosm model still need to be compared and validated with the actual situation in the natural environment to ensure accuracy and reliability. Our future studies will also continue to focus on the correlations and differences between the indoor model and natural conditions, and work to further improve the model conditions in the expectation of approximating natural lakes.
The residual concentration of florfenicol in nature can reach up to 11 mg/L. In our previous study, we found that the addition of florfenicol at 10 mg/L significantly suppressed the diversity of water denitrifying microorganisms and induced the accumulation of nitrate nitrogen and ammonium nitrogen in the aquatic microcosm. We add information about the concentration of florfenicol in different natural environments in line 47-54, thanks again.
-L.74... the effect of stress induced by florfenicol should have been supplemented with additional information regarding e.g. MIC MBC, which will have major impacts on the abundance and qualitative composition of the bacterial community.
>>>> Thanks very much for the reviewer’s valuable comments and suggestions. We added information on the MIC of florfenicol in the Materials and Methods section. The following information was added to manuscript " The in vitro bacteriostatic effect of the configured florfenicol solution was tested using the micro broth two-fold dilution method, and the MIC of florfenicol against Escherichia coli was found to be 8 μg/mL, which ensured that the solution had a significant bacteriostatic effect", thanks again.
-L.75… The goal is unconvincingly worded. The expected main contribution of the so called "novelty" of the study is not specified.
>>>> Thanks very much for the reviewer’s valuable comments and suggestions. In order to avoid to overstatement, we have revised the statement " and to provide a theoretical basis for the scientific use of florfenicol in aquaculture and the maintenance of a healthy aquaculture ecosystem ", and change it to“in sediment to florfenicol stress, and to provide an important basis for an in-depth understanding about the effects of florfenicol on the environmental microbial com-munity, and to provide the basis for better assessment and management regarding the environmental risk of florfenicol in the environment", thanks again.
-L. 98…how is the control marked?
>>>> Thanks very much for the reviewer’s valuable comments and suggestions. Our study included a time-series control group: 0 days and a control group of drug concentration: 0 mg/L, numbered DO and S0, respectively, and we added the relevant information in the Materials and Methods section in the manuscript, thanks again.
-L.103… these are speculations, unsubstantiated results. Have resistance genes been studied? L.103...
>>>> Thanks very much for the reviewer’s valuable comments and suggestions. This sentence is deleted here, and we put this sentence in the discussion to explore and speculate. In addition, we have conducted studies related to florfenicol resistance genes, but we did not analyze the two genes in association, and we will continue to investigate these aspects in subsequent studies, thanks again.
-L. 162... Fig.3 legend is not accurate. The colour range is bland.L. 162...
>>>> Thanks very much for the reviewer’s valuable comments and suggestions. We switched the type of Figure 3 from a heatmap to a more easily viewable bar chart with more distinct color differentiation, thanks again.
-L.172…. where specific results characterizing the claimed differences between the samples are given... "denitrification microbial community was more different between samples and more influenced by the drug"…. L.172....,
>>>> Thanks very much for the reviewer’s valuable comments and suggestions. We have added specific results for differences between samples here. The following formulation was added: “The treatment groups with higher florfenicol addition concentrations at 30 and 60 days were the furthest away from the control group, indicating that the nirS-type denitrification microbial community was more different and more influenced by the drug. Similarly, treatment groups with 10 and 100 mg/L florfenicol added concentrations at 30 and 60 days clustered together and were the furthest away from the control group in the cluster analysis. Samples with the same sampling time were clustered together in the cluster analysis. The cluster analysis results were consistent with the dispersion between samples in the PCoA analysis”, thanks again.
-L.223-225... these are unconvincing conclusions. Have similar results been found with other ATBs? Citations need to be added. Model studies have also been carried out, this may play a significant role. L.223-225......
>>>> Thanks very much for the reviewer’s valuable comments and suggestions. The effects of antibiotics on denitrification included key genes, functional enzyme activities, bacterial growth, and microbial community structure and diversity. In our study the addition of high concentrations of florfenicol resulted in a significant decrease in the diversity of nirS-type denitrifying microorganisms in sediments, but the abundance of the denitrifying gene nirS increased. This indicates that florfenicol affects sediment denitrification by influencing the growth of denitrifying bacteria and the structure and diversity of microbial communities. In our previous study, we found that the addition of florfenicol to aquatic microcosms suppressed the diversity of denitrifying microbial communities but increased the abundance of the nirS gene. We have added the results of the previous study to the paper, thanks again.
-L. 245-248…. The final conclusion is contradictory. The specific contribution of the study is not clearly stated. The expected aim of the study, that the results provide a theoretical basis for the scientific use of florfenicol in aquaculture and the maintenance of a healthy aquaculture ecosystem....was not fulfilled, as it is not clear how florfenicol will realistically affect the natural processes of microbial denitrification in the natural environment. L. 245-248....
>>>> Thanks very much for the reviewer’s valuable comments and suggestions. Firstly, in order to avoid to overstatement, we have revised the discussion in the manuscript and added a conclusion section in the manuscript, so that make the conclusion more specific and also deleted the statement " and to provide a theoretical basis for the scientific use of florfenicol in aquaculture and the maintenance of a healthy aquaculture ecosystem", thanks again.
-L. 267-277… Design of the experiment is insufficiently clarified.
Why were concentrations of the florfenicol in the decimal series chosen? Is it not justified how the sampling intervals were chosen? Is florfenicol's natural half-life interval not listed? Are control samples anywhere detail characterized? Has a positive check been made with, for example, another ATB? L. 267-277...
>>>> Thanks very much for the reviewer’s valuable comments and suggestions. Several florfenicol concentrations of 0, 0.1, 01, 10, and 100 mg/L were selected in the present study. Actually, our preliminary experiments showed that florfenicol 1 mg/L affected the number of microorganisms in aquatic microcosm. Therefore,1mg/L was selected as the medium dose group, and the low-concentration group and high-concentration group were set according to a 10-fold difference. This study is an indoor control experiment, with the aim of understanding how much higher the concentration of florfenicol affects the microbial diversity of aquatic microcosm, while how much lower the concentration of florfenicol has no significant effect on the microbial diversity of aquatic microcosm. Compared with the highest residual concentration of 11 mg/L reported in the literature, the designed concentration is either higher or lower than the concentration in the literature. Based on the results of the impact of different concentrations on environmental microbial diversity, the environmental safety of florfenicol residue can be evaluated.
The general rapid degradation phase of a drug is one week, so we chose 7 days as the first sampling time to assess the initial effect of the drug on the microbial community in the short term. The sampling time points of days 30 and 60 were used to assess the persistence and cumulative effects of the drugs. These time periods allow the drug to accumulate or break down in the aquatic microcosm, resulting in longer-term effects on sediment and biological systems, thanks again.
Forfenicol is stable physicochemically and persists in aquatic systems under ambient temperature and pH conditions typical of natural waters. Shannon M. Mitchell et al. 2015, published in Chemosphere, "Hydrolysis of amphenicol and macrolide antibiotics. Chloramphenicol, florfenicol, spiramycin, and tylosin" showed that florfenicol is relatively stable in water under ambient conditions, no degradation of florfenicol was observed at 25°C, and changes in pH did not significantly affect the degradation rate. Estimated half-lives in the pH 7 buffer at 60 °C were 38.1 d for florfenicol.
In addition, we have added information about the control group to make the control group more clear. It is regrettable that we did not use another antibiotic as a positive control. We will make improvements on these deficiencies in our future studies, thanks again.
-L- 279... devices are missing the manufacturer's indication.
>>>> Thanks very much for the reviewer’s valuable comments and suggestions. We have added relevant information in the manuscript, thanks again.
>>>>In addition to these revisions suggested by reviewer, we also improved our manuscript by supporting some other changes which have marked in red color in the revised manuscript. We hope these changes and corrections could make this manuscript more precise.
>>>>We have tried our best to improve the manuscript and made some changes in the manuscript that were marked. These changes will not influence the framework of the manuscript. We appreciate for Reviewer’s warm work earnestly, and hope that the corrections will meet with approval.